

# Evaluating a reservoir parametrisation in the vector-based global routing model mizuRoute (v2.0.1) for Earth System Model coupling

Inne Vanderkelen[1], Shervan Gharari[2], Naoki Mizukami[3], Martyn Clark[2], David M. Lawrence[3], Sean Swenson[3], Yadu Pokhrel[4], Naota Hanasaki[5], Ann van Griensven[1], and Wim Thiery[1]

[1]Vrije Universiteit Brussel, Department of Hydrology and Hydraulic Engineering, Brussels, Belgium
[2]University of Saskatchewan, Centre for Hydrology and Coldwater Laboratory, Canmore, Canada
[3]National Center for Atmospheric Research, Boulder, Colorado, USA
[4]Michigan State University, Department of Civil and Environmental Engineering, East Lansing, MI, United States
[5]National Institute for Environmental Studies, Tsukuba, Japan

**Correspondence:** Inne Vanderkelen (inne.vanderkelen@vub.be)

**Abstract.** Human-controlled reservoirs have a large influence on the global water cycle. While global hydrological models use generic parametrisations to model human dam operations, the representation of reservoir regulation is often still lacking in Earth System Models. Here we implement and evaluate a widely used reservoir parametrisation in the global river routing model mizuRoute, which operates on a vector-based river network resolving individual lakes and reservoirs, and which is currently being coupled to an Earth System Model. We develop an approach to determine the downstream area over which to aggregate irrigation water demand per reservoir. The implementation of managed reservoirs is evaluated by comparing to simulations ignoring inland waters, and simulations with reservoirs represented as natural lakes, using (i) local simulations for 26 individual reservoirs driven by observed inflows, and (ii) global-scale simulations driven by runoff from the Community Land Model. The local simulations show a clear added value of the reservoir parametrisation, especially for simulating storage for large reservoirs with a multi-year storage capacity. In the global-scale application, the implementation of reservoirs shows an improvement in outflow and storage compared to the no-reservoir simulation, but compared to the natural lake parametrisation, an overall similar performance is found. This lack of impact could be attributed to biases in simulated river discharge, mainly originating from biases in simulated runoff from the Community Land Model. Finally, the comparison of modelled monthly streamflow indices against observations highlights that the inclusion of dam operations improves the streamflow simulation compared to ignoring lakes and reservoirs. This study overall underlines the need to further develop and test water management parametrisations, as well as to improve runoff simulations for advancing the representation of anthropogenic interference with the terrestrial water cycle in Earth System Models.

## 1 Introduction

The terrestrial global water cycle is fundamentally altered by human activities like groundwater pumping, river water abstraction for irrigation and the construction of large dams (Oki and Kanae, 2006; Rockström et al., 2009; Wada et al., 2014). Worldwide, more than 45,000 large dams have been built, and the reservoirs thereby created provide hydropower, irrigation or



drinking water supply or are used for flood control (Lehner et al., 2011a; Sterl et al., 2020). Reservoir expansion since the $20^{th}$ century impounded at least 8,300 km$^3$ of water (Chao et al., 2008), counteracting global sea level rise by around 30 mm (Chao et al., 2008; Frederikse et al., 2020) and redistributing heat contained within the world's water resources and increasing anthro-
pogenic heat uptake by inland waters (Vanderkelen et al., 2020). By buffering seasonal river flow, reservoirs control more than half of the variability in global surface water storage (Cooley et al., 2021) and can substantially alter the timing, volume and the overall hydrograph of natural streamflow (Döll et al., 2009). Today, more than 77% of global rivers are human-regulated or are interrupted by dams, reservoirs or other infrastructure (Grill et al., 2019). Therefore, accounting for reservoirs and dam operations is important when assessing seasonality of global streamflow and water availability (Nazemi and Wheater, 2015;
Pokhrel et al., 2016).

Despite the clear human imprint on the terrestrial water cycle, Earth System Models generally do not yet account for human flow alterations by dam operations in their land component models (Pokhrel et al., 2016). Very recent efforts are aiming to address this limitation. For example, Zhou et al. (2020) coupled the MOSART-WM, a river routing and water management
model including reservoir operation, to the land model of E3SM. Also in MIROC-INTEG-LAND, water management modules have recently been incorporated in the land component of the MIROC Earth System Model, together with crop production, land ecosystem and land use modules (Yokohata et al., 2020). These developments overall suggest that reservoir management could potentially be considered in upcoming rounds of the Coupled Model Intercomparison Project (CMIP; Eyring et al., 2016) or other multi-model assessments.

Due to their importance for water resource assessments, reservoir operations have long since been represented in large-scale hydrological models, including catchment models (e.g. Chawanda et al., 2020; Shin et al., 2019), water management models (e.g. Voisin et al., 2013b, a) and Global Hydrological Models (GHMs; see Sood and Smakhtin (2015) and Telteu et al. (2021) for a comprehensive overview). However, substantial variations in operating rules and the lack of operational knowledge of
reservoirs worldwide, necessitate the use of generic parametrisations to describe reservoir operations (Pokhrel et al., 2016). Such generic schemes are typically not designed to reproduce the daily operations at individual reservoirs, but provide simple, yet widely applicable rules, mimicking human decisions in regulating dams, to the extent possible. A wide range of approaches exist, which can broadly be categorised into optimization-based methods (e.g. Haddeland et al., 2006), methods based on target storage and release (e.g. Burek et al., 2013; Yassin et al., 2019) and inflow-and-demand-based methods (e.g. Wisser et al.,
2010; Hanasaki et al., 2006). In addition to these approaches, which do not require prior information on historical reservoir operations, there are also a wide variety of reservoir models that use operational data for specific reservoirs to develop general operational rules (e.g. Coerver et al., 2018; Zhao et al., 2016; Ehsani et al., 2016). For an elaborated overview of the range of existing reservoir parametrisations, their characteristics, advantages and disadvantages, the reader is referred to Pokhrel et al. (2016); Yassin et al. (2019) and Gutenson et al. (2020).






Here, we evaluate the representation of reservoirs in the state-of-the art river routing model mizuRoute (Mizukami et al., 2016, 2021), in view of its anticipated coupling in the Community Land Model (CLM), the land component of the Community Earth System Model (CESM). The CLM modelling framework already accounts for historical reservoir construction by including lake area expansion (Vanderkelen et al., 2021), but an explicit representation of lake and reservoir water balance dynamics is currently lacking. We investigate the effect of dam operations on river flow when using the parametrisation of Hanasaki et al. (2006) in mizuRoute. Compared to other reservoir models, this parametrisation has low data requirements (only information on irrigation water demand and instantaneous inflow is needed), does not require prior knowledge (e.g. on future inflows, like the schemes derived from the Haddeland et al. (2006) parametrisation) and can thus be used instantaneously during a simulation. Moreover, due to its generic nature, it can be applied to every reservoir across the globe. Therefore, this parametrisation has been widely used as a basis in large scale hydrological modelling studies (e.g. Biemans et al., 2011; Voisin et al., 2013a; Droppers et al., 2020; Döll et al., 2009; Hanasaki et al., 2008; Pokhrel et al., 2012; Shin et al., 2019).

In contrast to previous studies, we evaluate the implementation of the Hanasaki et al. (2006) parametrisation in a global river routing model that operates on a vector-based river network, mizuRoute. To provide seasonal irrigation demand per reservoir, we develop an irrigation topology, which defines the area over which the water demand is aggregated for an individual reservoir, based on the river network topology and catchments. We evaluate the added value of the Hanasaki et al. (2006) parametrisation for reservoir outflow and storage modelling in a stand-alone mizuRoute simulation that uses reservoir observations as input, and compare results to a simulation using the natural lake outflow parametrisation of Döll et al. (2003). Next, both parametrisations are evaluated using mizuRoute in a global routing-only application with runoff input from CLM for their ability to represent outflow and storage at individual reservoirs and to capture long-term trends in monthly streamflow indices. Our modelling framework allows us to identify biases in runoff from CLM by comparing previously not modelled variables (e.g. reservoir outflow and storage) to observations. Finally, we explore new avenues for future model development and towards coupling within CESM. This study provides an essential step towards the incorporation of human water management and reservoir dynamics in a coupled model, which will enable to investigate the climate change impacts on human water management and the potential of water management strategies to mitigate climate change impacts on water resources.

## 2 Modelling framework

### 2.1 mizuRoute

The vector-based routing model mizuRoute is designed to process runoff provided by hydrological models or land models to spatially distributed river streamflow (Mizukami et al., 2016). The routing is performed in two steps: first, basin runoff is routed from the hillslopes to the river reach with a gamma distribution based unit-hydrograph. Then, the water is routed downstream through the river channel network, using either an impulse response function (IRF) or a kinematic wave tracking (KWT) routing scheme (Mizukami et al., 2016). In stand-alone applications, mizuRoute internally remaps the gridded runoff provided by the land model or hydrological model to the basin defined in the vector-based river network. In continental or global applications,



mizuRoute can operate with decomposed river networks to allow for parallel routing computations (Mizukami et al., 2021).
Natural lakes and reservoirs are integrated in the vector-based river network as hydrological features with additional parameters including information on the characteristics of the lake and/or reservoir, like maximum capacity (Gharari et al., in prep.). This approach allows to model the lake and reservoir water balance, using data on precipitation and evaporation from the water surface, in combination with parametrisations providing information on the releases, both natural outflows and regulated discharge. For this study, the IRF routing scheme was used for river channel routing that produces the discharge into lakes and
reservoirs.

## 2.2   Lake and reservoir parametrisations

Gharari et al. (in prep.) introduces parametric lake and reservoir implementations in mizuRoute to model lake and reservoir outflow. Natural lakes are modelled as linear reservoirs using the parametrisation of Döll et al. (2003) (eq. 1) which resolves daily outflow ($Q_{daily}$ in m$^3$ s$^{-1}$), as a function of current active lake storage ($S$ in m$^3$) with a release coefficient $k_r$ (taken
constant at 0.01 s$^{-1}$) and the maximal lake storage capacity ($S_{max}$ in m$^3$). The exponent in the parametrisation is determined based on the theoretical value of outflow over a rectangular weir (Meigh et al., 1999).

$$Q_{daily} = k_r \cdot S \cdot \left( \frac{S}{S_{max}} \right)^{1.5} \tag{1}$$

In this study, we investigate the impact of implementing management of human-constructed reservoirs and dam-controlled lakes with the parametrisation described in Hanasaki et al. (2006). This algorithm targets minimization of intra- and inter-annual
variability, while accounting for irrigation and other water demands, making a distinction between reservoirs used for irrigation and other purposes such as hydropower, flood control, navigation or water supply. Irrigation reservoirs, which provide water for crops downstream, are characterised by a distinct seasonal variability guided by the downstream irrigation water needs. Since withdrawal periods do not necessarily coincide with high inflow periods, the parametrisation explicitly takes into account the downstream irrigation demand in the intra-annual outflow. The reservoirs with purposes other than irrigation are operated in
the same way, aiming to reduce intra- and interannual flow variability. Furthermore, the parametrisation differentiates between "multi-year reservoirs" with high storage capacity compared to their annual inflow, and "within-a-year reservoirs", defined as reservoirs with annual inflow values that are more than half of the storage capacity. "Within-a-year reservoirs" carry the inflow seasonality in their outflow values to compensate for potential overflow and storage depletion, while "multi-year reservoirs" aim to maintain a constant outflow (Hanasaki et al., 2006).


Below, we outline the parametrisation as described in Hanasaki et al. (2006) and specify how it is implemented in mizuRoute. The parametrisation uses operational years, which are unique to every reservoir and different from the calendar year. The operational year starts on the first day of the month in which the multi-year monthly inflow drops below the annual inflow (Hanasaki et al., 2006; Haddeland et al., 2006). Then, at the start of the operational year, the monthly target release is determined





based on the purpose of the reservoir. For non-irrigation reservoirs the monthly target release $Q_{target}$ (m$^3$ s$^{-1}$) is taken as the annual mean inflow $I_{mean}$ (m$^3$ s$^{-1}$; eq. 2).

$$Q_{target} = I_{mean} \qquad (2)$$

For irrigation reservoirs, the target release is calculated by equation 3,

$$Q_{target} = \begin{cases} 0.1 \cdot I_m + 0.9 \cdot I_{mean} \cdot \frac{D_m}{D_{mean}} & , \text{if } D_{mean} \geq \beta \cdot I_{mean} \\ I_{mean} + D_m - D_{mean} & , \text{otherwise} \end{cases} \qquad (3)$$

with $I_m$ (m$^3$ s$^{-1}$) the mean monthly inflow for the corresponding month, $I_{mean}$ (m$^3$ s$^{-1}$) the mean annual inflow, $D_m$ (m$^3$ s$^{-1}$) the mean monthly irrigation water demand for the corresponding month, $D_{mean}$ (m$^3$ s$^{-1}$) the mean annual irrigation demand and $\beta$, a coefficient representing the minimum release to meet environmental requirements (here $\beta$=0.9, leaving 10% of annual mean flow available to meet environmental requirements). Following the adjustments of Biemans et al. (2011) to the original Hanasaki et al. (2006) parametrisation, we only account for irrigation water withdrawal, while neglecting domestic and indus-

trial water use. In addition, we also apply a minimum environmental flow requirement of 10% of mean annual inflow, instead of 50% used by Hanasaki et al. (2006) to ensure enough water is retained in the reservoirs during low-flow months to meet the irrigation demands (Biemans et al., 2011).

The actual release depends on how full the reservoir is at the start of the operational year, determined by the release coefficient

($E_r$, eq. 4), giving the ratio between the reservoir storage at the start of the operational year ($S_{ini}$, m$^3$) and the maximal storage capacity ($S_{max}$, m$^3$), scaled with $\alpha$ (set constant at 0.85). This coefficient quantifies the share of the total storage that is considered active storage, i.e. total storage excluding dead and emergency storage.

$$E_r = \frac{S_{ini}}{\alpha \cdot S_{max}} \qquad (4)$$

The actual reservoir release ($Q_{daily}$, m$^3$ s$^{-1}$) depends on the reservoir type ("multi-year" or "within-a-year", defined by the

capacity ratio $c$ (given by $S_{max}/I_{mean}$), and is calculated by eq. 5.

$$Q_{daily} = \begin{cases} E_r \cdot Q_{target} & , \text{if } c \geq 0.5 \textit{ (multiyear reservoir)} \\ \left(\frac{c}{0.5}\right)^2 \cdot E_r \cdot Q_{target} + \left\{1 - \left(\frac{c}{0.5}\right)^2\right\} \cdot I_{daily} & , \text{if } c < 0.5 \textit{ (within-a-year reservoir)} \end{cases} \qquad (5)$$

In this study, we prescribe the seasonal cycles for monthly mean inflow and demand based on naturalized simulations, but the implementation allows for transitioning from prescribed values to modelled mean inflows and demands over the last 5 years, similar to the approach of Biemans et al. (2011); Droppers et al. (2020). Using time-varying inflows and demands allows





the model to respond to climatological changes when determining reservoir release, which is a capability that is particularly relevant in the context of climate change studies. When the reservoir storage drops below the dead storage level, defined as 10% of the maximal reservoir storage, no water is released. When the simulated storage exceeds the maximal reservoir capacity, the surplus is released as spillway overflow. Hence, the calculated reservoir release is required to be between these two constraints so as to keep reservoir storage within realistic limits.

## 2.3 Irrigation topology

The Hanasaki et al. (2006) parametrisation for irrigation reservoirs requires mean monthly irrigation water demand per reservoir as an input. Previous studies with grid-based river models defined the dependent area of a reservoir by number of cells downstream either to the next reservoir, the river mouth, a predefined maximum number of downstream cells (e.g. 5 cells at 0.5° or 10 cells at 1°, corresponding to the typical distance that river water travels within a month, Döll et al., 2009; Hanasaki et al., 2008), or grid cells which are located at a predefined threshold distance from the main river reach (e.g. 200 km or 2° Biemans et al., 2011; Voisin et al., 2013a). A vector-based river network, in contrast, needs a reservoir dependency database ('irrigation topology'), which provides for each reservoir the river segments and corresponding hydrological response units (HRUs) to which it supplies irrigation water. When multiple reservoirs serve the same HRU, the irrigation topology should also include the share of the different reservoirs in meeting the water demand of the individual HRU. The total water demand of a reservoir is then calculated by taking the weighted sum of the irrigation demands of HRUs, which are dependent on that specific reservoir.

Here, we develop a global irrigation topology based on simple rules, in line with other large-scale hydrological models. Our approach utilizes the topological relation provided in the vector-based river network topology as well as the bottom elevation of each HRU. First, the reservoir for which the calculations will be done is selected and the corresponding segment on the river network is localised. Then, the downstream river segment for which the reservoir influence ends is determined based on a distance threshold along the main stem (here taken at 700 km). If the river mouth or another reservoir is located within this distance threshold, their corresponding segments are chosen as the ending segment. All HRUs corresponding to the segments along the main river stem and first order tributaries are added to the dependency data set. Third, the HRUs of all higher order tributaries below a threshold river length from the main stem (here taken at 100 km), are added. Finally, the HRUs with higher bottom elevation than the reservoir segment are excluded, to avoid cases where irrigation water would be transported uphill. This HRU selection procedure is showcased for the Island Park reservoir of the Snake river basin in Fig. 1a. The selection routine is repeated for every reservoir in the river network. For HRUs with two or more dependent reservoirs (Fig. 1b), the demand is distributed among the reservoirs along their ratio of the maximum storage capacity, following the approach of Haddeland et al. (2006) and Voisin et al. (2013a). Finally, the irrigation topology is used to derive the total irrigation demand for every reservoir based on the HRU irrigation water demands for every time step (Fig. 1c).





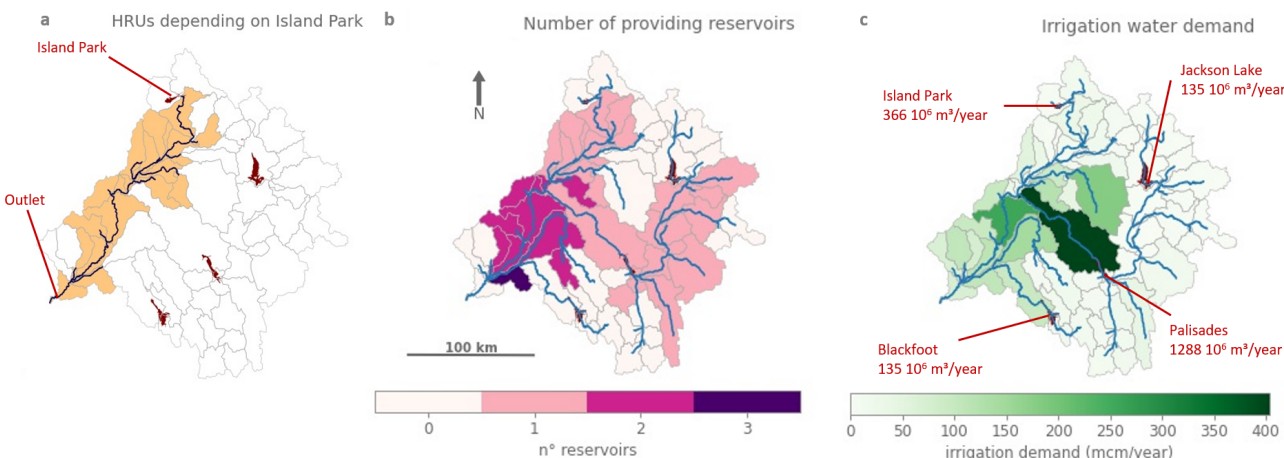

**Figure 1.** Illustration of the irrigation topology for the Snake River basin (with basin outlet taken at American Falls reservoir, ID, United States of America). Selection of river segments and corresponding downstream HRUs of the Island Park reservoir (panel a), number of reservoirs supplying water to each HRU (panel b), total irrigation water demand per HRU and reservoir, calculated using the irrigation topology (panel c). Reservoir locations from GRanD, river network from HDMA and irrigation demand remapped from a gridded CLM simulation (see section 3.2).

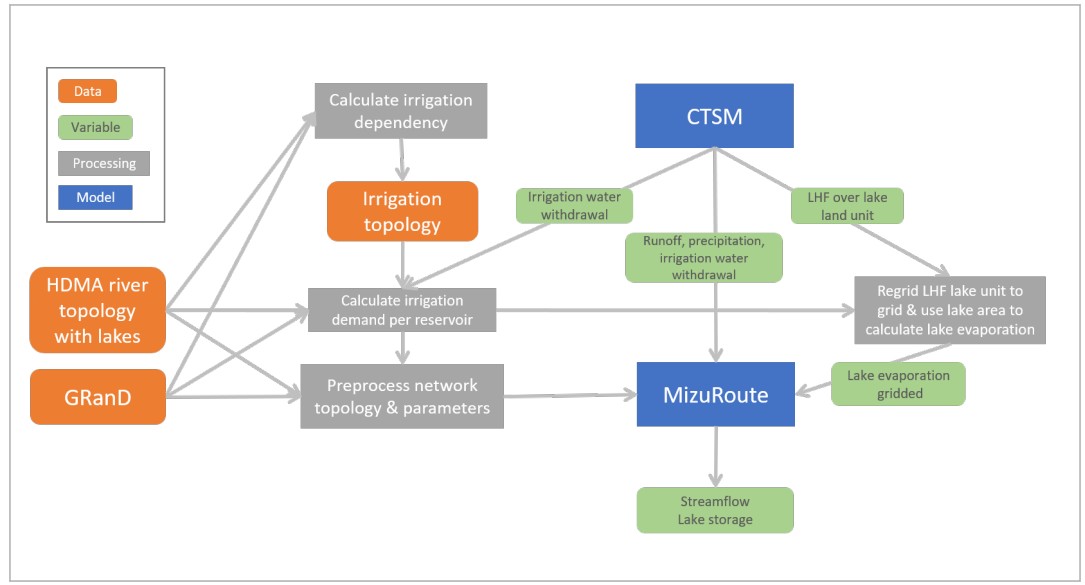

**Figure 2.** Schematic representation of the modelling workflow for the global-scale mizuRoute application, using input data and parameters based on Community Land Model (CLM) simulations. LHF refers to latent heat flux.



## 3    Simulation setup

The lake and reservoir parametrisations in mizuRoute are evaluated both in a local and global setting. By using observed
streamflow values as forcing, the local mizuRoute application allows for direct evaluation of the implementation of the different
outflow schemes. In the global-scale mizRoute application, outlined in Fig. 2, the reservoir schemes are embedded in global-
scale routing simulations that receive forcing fields directly from the land model.

### 3.1    River network topology

The Hydrologic Derivatives for Modeling and Applications (HDMA;  Verdin, 2017) is a vector-based river network based on
HydroSHEDS, GMTED2010 and SRTM DEMs and entails 295,335 river reaches and HRUs, with a scale of 25 km$^2$ (the mini-
mum upstream area to define the start of a river reach). Lakes are included on the HDMA river network by geo-referencing lake
polygons of the HydroLAKES dataset (Messager et al., 2016a) with a surface area larger than 10 km$^2$ to their corresponding
river reaches (Gharari et al., in prep). The lake polygons from the HydroLAKES dataset are linked to the Global Reservoir
and Dam dataset (GRanD;  Lehner et al., 2011a) which provides additional information about reservoirs including maximum
reservoir capacity and reservoir purpose. Based on this information, a lake segment is classified as a reservoir if it is present in
GRanD (including both man-made reservoirs and dam-controlled lakes). In total, 1773 reservoirs from GRanD are included in
the river network, of which 484 are categorized as irrigation reservoirs.

### 3.2    Land model forcing

We conducted global land-only simulation with the Community Land Model (CLM;  Lawrence et al., 2019) that receives pre-
scribed meteorological conditions from the Global Soil Wetness Project (GSWP3; http://hydro.iis.u-tokyo.ac.jp/GSWP3/ see
also Lawrence et al. (2019)) and prescribed vegetation phenology from MODIS (IHistClm5SP component set). The simulation
is run on a 0.5° by 0.5° grid, for the period 1961-2015 (including 5 years for spin up). The simulation is conducted with the
updated lake and reservoir mask based on HydroLAKES and GRanD as described in Vanderkelen et al. (2021) and the de-
fault irrigation algorithm, without constraints on water availability. Therefore, simulated grid cell irrigation water withdrawal
corresponds to the total irrigation water demand of the grid cell. The daily simulated gridded runoff is directly used as input
to mizuRoute. Furthermore, the precipitation and evaporation over lakes and reservoirs, necessary for their water balance, are
also provided by CLM and remapped within mizuRoute. Precipitation is directly provided, while lake evaporation is calculated
in an intermediate processing step, that is, by converting the latent heat flux at the lake 'land unit' level to evaporation using
the latent heat of vaporization (2.501 ·10$^6$ J kg$^{-1}$).






### 3.3 Parameters of the outflow parametrisations

All parameters required for the lake and reservoir schemes are provided through the network topology (Appendix Table A1). Maximum reservoir capacity and the reservoir purpose are both provided by the attributes from GRanD. Only the reservoirs for which GRanD assigns irrigation as the main purpose are categorized as irrigation reservoirs in mizuRoute. At the start of the simulation, the initial storage is set at the maximal storage capacity. In the local mizuRoute simulations, monthly mean inflow values are calculated based on observed inflows according to their availability (Appendix Table A1). For the global-scale mizuRoute simulations, monthly mean inflow values per reservoir are obtained from a mizuRoute simulation with only natural lakes using the Döll et al. (2003) parametrisation for the period 1979-2000. For both the local- and global-scale simulations, mean monthly irrigation water demands per reservoir are calculated based on the gridded CLM simulation for the same period. The gridded demands are first remapped to the HRUs of the vector-based river network, and subsequently the irrigation topology described in section 2.3 is applied, using dependency thresholds of 700 km (maximum downstream distance along the main river stem), and 100 km (maximum distance along tributaries from the main river stem).

### 3.4 mizuRoute simulations

First, local mizuRoute simulations are conducted for 26 individual reservoirs, using observed reservoir inflows as input forcing (section 4.1, appendix table A2). Reservoir outflow is either modelled as a natural lake with the Döll et al. (2003) parametrisation (hereafter denoted as NAT), as a human-operated reservoir with the Hanasaki et al. (2006) parametrisation (DAM) or as run-of-the river assuming there is no reservoir, using observed inflow as outflow (NOLAKES). To evaluate the use of the Hanasaki et al. (2006) parametrisation for irrigation reservoirs in particular, additional simulations are conducted with all reservoirs considered as non-irrigation reservoirs (DAM_NOIRR). Simulations are performed at daily time step, but compared to observations according to the observational time steps (daily for 18 reservoirs and monthly for 8 reservoirs).

Second, four global-scale mizuRoute simulations are conducted on a daily time step using the HDMA river network topology, gridded runoff from CLM and the IRF-UH routing method. Similar to the local simulations, four simulation types are performed. The first uses the Döll et al. (2003) parametrisation for all reservoirs and lakes on the river network (NAT). The second simulation (DAM) uses the parametrisation of Hanasaki et al. (2006) for reservoirs and dam-controlled lakes, in addition to Döll et al. (2003) for the natural lakes. Third, all lakes and reservoirs are treated as normal river segments (NOLAKES). Finally, an additional simulation is performed, similar to DAM but with all reservoirs considered as non-irrigation reservoirs (DAM_NOIRR). Comparing this simulation to the DAM simulation allows to asses the added value of accounting for irrigation water demand using our irrigation topology. Every simulation is conducted for the period 1979-2000, of which the two first years are considered spin up and are excluded from the analysis.





## 4 Evaluation data sets and metrics

Both local and global-scale mizuRoute simulations are evaluated with observations of 26 individual reservoirs. In addition, the global-scale mizuRoute simulations are compared to global streamflow indices.

### 4.1 Local reservoir observations

Observations for reservoir inflow, outflow and storage are retrieved from the data set of Yassin et al. (2019), including information on 37 reservoirs worldwide assembled from different sources. We use a subset of 26 reservoirs from this dataset, corresponding to the reservoirs that could be located on the HDMA river network, and thus are modelled in our mizuRoute simulations (Table A2). Due to data availability, these reservoir observations are not evenly distributed over the globe (Fig. 6). The dataset provides daily inflow, storage and outflow observations for 18 reservoirs, and monthly observations for the

remaining 6 reservoirs.

### 4.2 Global streamflow indices: observations from GSIM

The Global Streamflow Indices and Metadata archive (GSIM) is a worldwide collection of indices derived from more than 35,000 daily streamflow time series (Do et al., 2018). The dataset provides quality controlled time series indices on yearly, seasonal and monthly resolution compiled from 12 databases with daily streamflow, including both research databases and

national databases (Do et al., 2018; Gudmundsson et al., 2018a). Here, we use the following indices, all on a monthly time scale: mean daily streamflow (MEAN; $m^3$ $s^{-1}$), standard deviation of daily streamflow (SD; $m^3$ $s^{-1}$) and the minimum and maximum daily streamflow (MIN and MAX, $m^3$ $s^{-1}$). We only use stations that are located on the river network, based on the coordinates of the stations. First, the stations with suspect coordinates are excluded. Then, we select all stations with observation periods overlapping the simulations period (1981-2000) and within a 0.002° spatial error tolerance limit on the

river network (10,233 stations). Finally, only stations less than 200 km downstream of a simulated reservoir are kept. This is results in 406 GSIM stations used in the analysis.

### 4.3 Global G-RUN runoff reconstructions

We evaluate CLM runoff using the global runoff reconstruction from the G-RUN ENSEMBLE (Ghiggi et al., 2019, 2021). G-RUN provides monthly runoff rates on a 0.5° grid for 1971-2010, based on upscaled river discharge using a machine learning

algorithm (Ghiggi et al., 2019). The G-RUN ENSEMBLE extends the original G-RUN based on GSWP3 with 21 different atmospheric datasets (Ghiggi et al., 2021). In this study, we use the ensemble mean averaged for 1971-2000.

### 4.4 Evaluation metrics

Simulated time series are compared to observations for their corresponding periods using the Kling Gupta Efficiency (KGE; Gupta et al., 2009) and the absolute percent bias (PBIAS; eq. 6).





$$|PBIAS| = \frac{\sum_{i=1}^{n} |m_i - o_i|}{\sum_{i=1}^{n} o_i} \tag{6}$$

with $n$ the number of observations $m$ and $o$ the simulated and observed series, respectively. To investigate the role of the different components, we use the terms of the decomposed KGE following eq. 7 (Gupta et al., 2009).

$$KGE = 1 - \sqrt{(r-1)^2 + \left(\frac{\sigma_{mod}}{\sigma_{obs}} - 1\right)^2 + \left(\frac{\mu_{mod}}{\mu_{obs}} - 1\right)^2} \tag{7}$$

with $r$ the linear correlation between simulated and observed values, $\frac{\sigma_{mod}}{\sigma_{obs}}$, the ratio of modelled and observed standard

deviation, representing the variability error, and $\frac{\mu_{mod}}{\mu_{obs}}$, the ratio of the modelled and observed means, representing the mean bias. Following Knoben et al. (2019), KGE values above -0.41 are considered good model performance compared to the mean flow benchmark.

## 5 Results

### 5.1 Local mizuRoute simulations

The local mizuRoute simulations with observed daily reservoir inflows allow to directly compare the different outflow parametrisations and run-of-the river conditions (Fig. 3). For outflow, the DAM simulation produces the highest KGE scores for 12 of 26 reservoirs (Fig. 3a), while the NAT simulation performs best for 8 reservoirs. The NOLAKES simulation typically yields good skill for reservoirs with a low capacity ratio, where outflows are strongly influenced by inflow seasonality as their storage capacity is small compared to the annual mean inflow (upper half of Fig. 3a, appendix Figs. A2 and A4). For all simulations,

the performance of simulated outflow decreases with increasing reservoir capacity ratio, apart from a few exceptions.

For storage, the DAM simulation outperforms NAT for most reservoirs (18 out of 26; Fig. 3b, appendix Figs. A2 and A4), with a median KGE of 0.4 compared to 0.08. Especially for reservoirs with a high capacity ratio, DAM shows notably higher KGE values compared to NAT. This demonstrates the added value of the Hanasaki et al. (2006) parametrisation in minimiz-

ing the inter-annual outflow variability for reservoirs with a high capacity ratio. The individual time series of modelled storage show systematic over- and underestimation for the Glen Canyon, Amistad and Navajo reservoirs, with excessive outflow values indicating the reservoir reached its maximum capacity (appendix Fig. A2 and A4). In our modelling workflow, the maximum storage capacity is provided by GRanD for all reservoirs on the river network, and therefore these systematic storage biases may be caused by discrepancies between the real reservoir capacity and those reported in GRanD (e.g. for the Navajo reservoir,

GRanD reports a maximum capacity of $1278 \cdot 10^6$ m$^3$, while the US Board of Reclamation reports a capacity of $2107 \cdot 10^6$ m$^3$, which better corresponds to the observations).





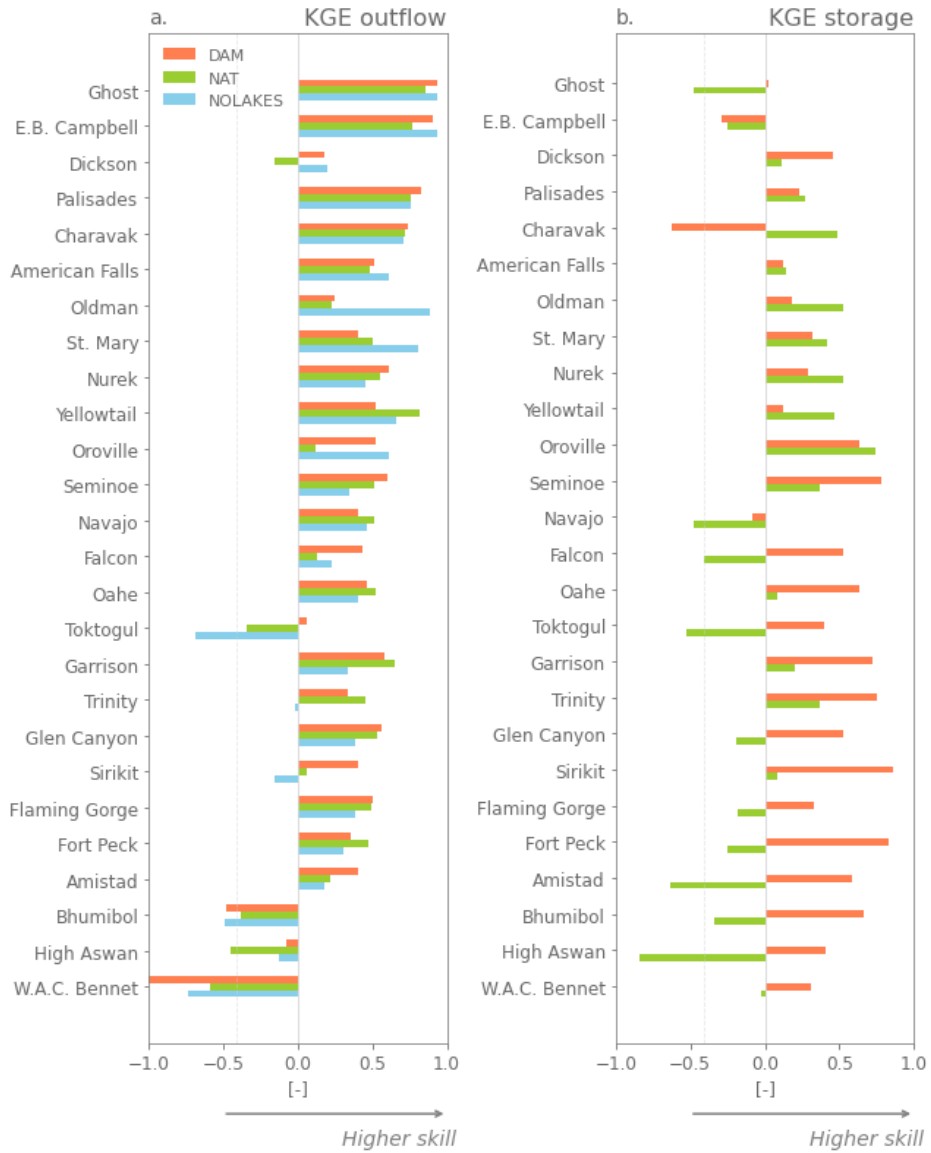

**Figure 3.** Evaluation using Kling-Gupta Efficiency (KGE) of the Hanasaki et al. (2006) (DAM) and (Döll et al., 2003) (NAT) parametrisations with observed inflows, and using inflow as outflow (assuming there is no lake; NOLAKE) against observed outflow (panel a) and observed storage (panel b) using observations from Yassin et al. (2019). The reservoirs are ordered from low to high capacity ratio, defined as the ratio between the mean annual inflow and storage capacity (see Table A2).

The comparison of the DAM with the DAM_NOIRR simulations for irrigation reservoirs reveals that accounting for irrigation only has a limited effect in the current implementation, except for Oldman, St-Mary, Nurek, Sirikit and Bhumibol reservoirs, where accounting for irrigation demands improves the outflow simulation (appendix Figs. A1 and A2). For ex-






ample, Bhumibol and Sirikit are multi-year irrigation reservoirs with a clear irrigation signature in their observed outflow seasonality, as they buffer water during the high-flow season to release for irrigation during the low-flow season (Hanasaki et al., 2006). The simulated annual outflow cycle for the Sirikit reservoir shows slightly increased outflows during the low-flow season (February-May) for the original Hanasaki parametrisation compared to Hanasaki without irrigation demands (Fig. A2).

The limited added value of accounting for irrigation demands for the 12 irrigation reservoirs suggests that reservoir irrigation demands are likely underestimated in this modelling framework (see section 6.2).

## 5.2 Global-scale mizuRoute simulations: evaluation with reservoir observations

As expected, the global-scale mizuRoute simulations show overall substantially lower performance at the evaluation sites compared to the simulations using observed inflows, with median KGE values for outflow of -0.29, -0.29 and -0.35 for the DAM,

NAT and NOLAKES simulations, respectively (Fig. 4b). Most reservoirs have negative KGE scores, and for four out of 26 reservoirs all simulations are outperformed by the mean annual flow benchmark. In terms of percent absolute bias for outflow, the difference between the DAM and NAT is very small or negligible for more than half of the reservoirs (Fig. 4a). This is also visible in the small differences between simulations in the bias term of KGE, in particular for DAM and NAT (Fig. A4b). For correlation, the NAT simulation has the best skill for 15 of 21 reservoirs, with highest correlations for reservoirs with low

capacity ratios (Fig. A4c). The added value of using the Hanasaki et al. (2006) parametrisation for reservoir storage is less apparent in the global-scale mizuRoute simulation, as the DAM simulation outperforms the NAT simulation for 10 of the 21 reservoirs for absolute percent bias and KGE (Fig. 5). Consistent with the observation-driven local simulations, the global-scale DAM simulation performs systematically better for reservoirs with a high capacity ratio, and in most cases better than NAT.

While in the local mizuRoute application, the DAM simulation outperforms the NAT and NOLAKES simulations for most reservoirs, especially for storage, this is not the case in the global-scale mizuRoute simulations. The main cause for these discrepancies are biases in the simulated reservoir inflow, which could be originating from biases in the simulated runoff from CLM or from small reservoirs upstream and their dam operations which are not resolved in the HDMA river network, with the resultant streamflow alterations not included in the river flow. For 15 of the 21 reservoirs in the dataset, however, there is

at least one upstream reservoir resolved in the HDMA river network, as only 6 reservoirs have no upstream reservoir resolved (Trinity, Navajo, Oldman, Seminoe, Sirikit and St-Mary). The next section therefore focuses on the biases in simulated inflow and runoff.





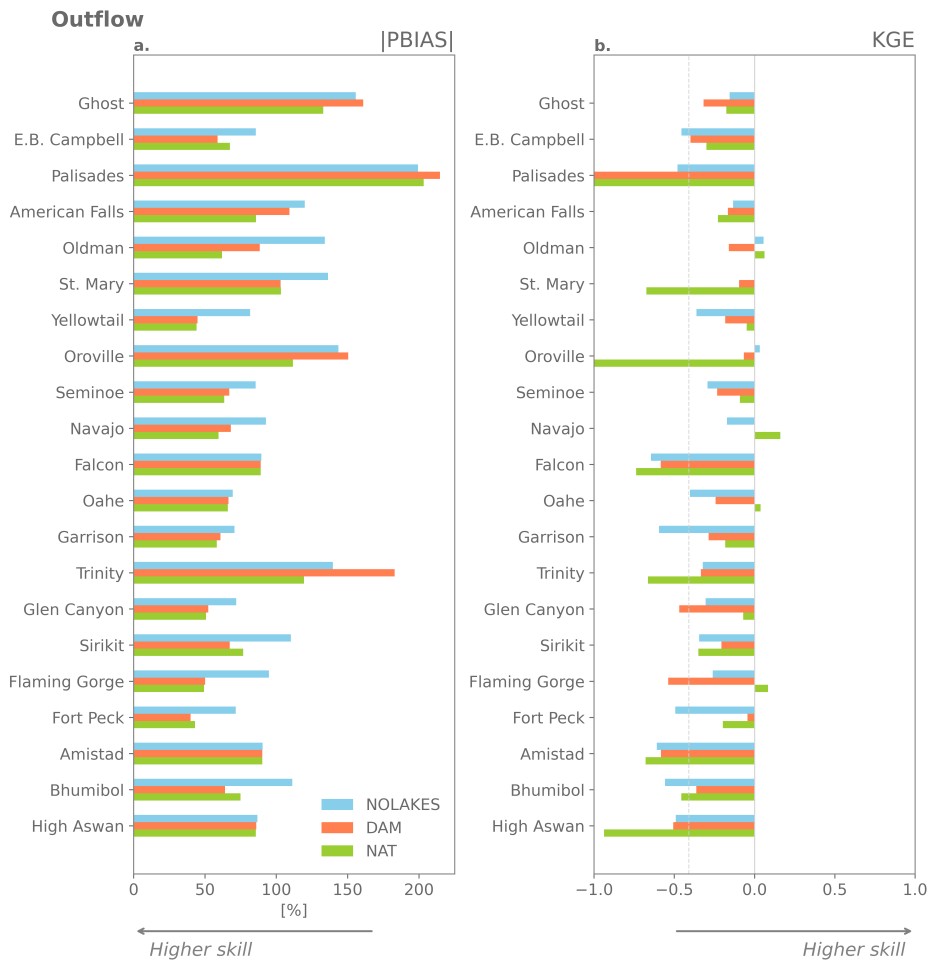

**Figure 4.** Performance of the global-scale mizuRoute simulations for outflow compared to reservoir observations using absolute percent bias (|PBIAS|; panel a) and Kling-Gupta Efficiency (KGE; panel b)





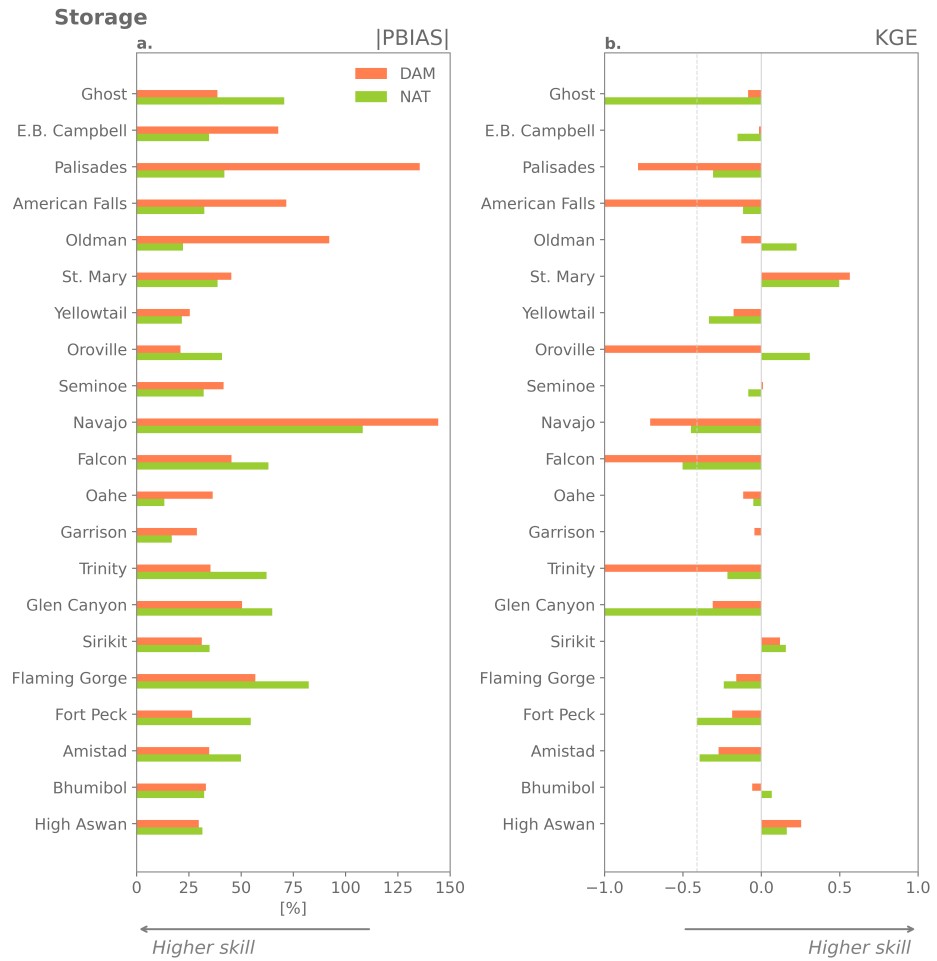

**Figure 5.** Performance of the global-scale mizuRoute simulations for storage compared to reservoir observations using the absolute percent bias (|PBIAS|; panel a) and Kling-Gupta Efficiency (KGE; panel b).





### 5.3 Inflow and runoff bias of CLM forcing

The comparison of simulated spatially distributed runoff from CLM with the global reconstructions of G-RUN uncovers sub-
stantial biases (Fig. 6). The mean annual runoff bias is +0.077 mm day$^{-1}$, but regionally large differences exist: runoff is
overestimated in the northwestern Amazon, West Africa, large parts of China, West India, Japan, and to a lesser extent in
central US and the European mainland. CLM underestimates runoff in the tropical rainforest areas of central Amazonia and
the Congo basin, and in mountain areas, like the Pakistani mountain ranges, the European Alps, the Rocky Mountains in the
US and Canada, the northern part of the Andes and the Southern Alps in New Zealand.


As 20 reservoirs in the dataset are located in the Central and Western parts of the Contiguous United States and Canada,
we focus on these regions to compare runoff and reservoir inflow seasonality to observations (Fig. 7). In the plains, runoff is
generally slightly overestimated, while in the mountainous areas like the Rocky Mountains, Sierra Nevada and Cascade Range
mean annual runoff is substantially underestimated (Fig. 7a). Via flow routing, these runoff biases translate into streamflow
biases (Fig. 7b-s).

Overall, the simulated streamflow deviates from the observed seasonal cycles in terms of absolute bias, timing of the high
flows and amplitude. The deviations can thereby roughly be grouped in four categories of reservoirs. First, for large reservoirs
like Amistad and Falcon International on the Rio Grande, and Garrison and Oahe on the Missouri river (Fig. 7h-k), mizuRoute
(forced with CLM output) largely overestimates the observed inflows (up to +1434 % for Falcon). For these reservoirs the
upstream flows are highly regulated by dam operations and the positive inflow biases are therefore likely originating from
unrepresented upstream dam operations (Shin et al., 2019) or from positive biases in simulated runoff (see discussion section
6.1). Other reservoirs have inflows highly controlled by snow melt, with their headwaters in the Rocky Mountains (Flaming
Gorge, Navajo, Palisades, American Falls and Glen Canyon reservoir; Fig. 7n, l, r, q, m). For most of these reservoirs, the
annual peak in inflow, likely coming from snow melt, is simulated 2-3 months too early (March-April-May) compared to the
peak in observed inflows (June-July-August). This is also the case for the small within-a-year reservoirs in the Canadian Rocky
Mountains (Oldman, Saint-Mary and Ghost reservoirs; Fig. 7b, c, s). These biases in runoff timing could potentially be related
to unresolved topography in these coarse resolution simulations. For the mainly rain-fed Oroville and Trinity reservoirs (Fig.
7o, p), the release period is simulated too early in the year. Finally, some Canadian reservoirs, like E.B. Campbell show only
little variation in storage, which could in part be explained by the linkages of these reservoirs with lake and swamp systems.

These inflow discrepancies point at deficiencies in the simulated runoff, as the comparison of spatially aggregated runoff
from CLM versus G-RUN over the reservoir catchments show similar patterns (not shown). Moreover, local systematic biases
in runoff are aggregated over the catchment and result in magnified inflow biases. Previous research showed that the runoff in-
puts are a more important bias source for river discharge in mizuRoute compared to the river network and routing scheme when
analysed on monthly time steps (Mizukami et al., 2021). Also in other large-scale hydrological models, annual river discharge





show broad range of values and large differences in runoff ratios among different models (Masaki et al., 2017; Haddeland et al., 2011).

The inflow biases are adversely affecting the skill of the reservoir parametrisation in the global-scale mizuRoute simulations compared to the local applications, especially for reservoir storage. We therefore anticipate that when the runoff simulations are improved within the driving land model, in this case CLM, improved results can be expected also in global-scale mizuRoute simulations. Therefore, we focus on comparing the DAM simulation to the NOLAKES simulation in the remainder of this paper.

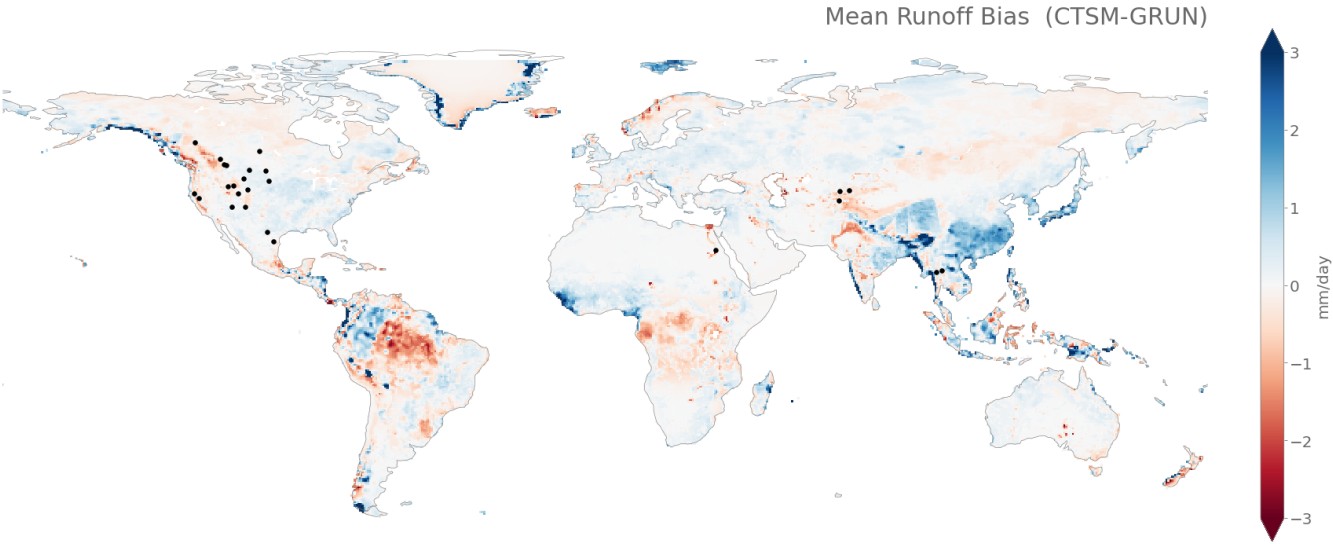

**Figure 6.** Mean runoff bias of CLM, compared to G-RUN for the period 1971-2000. Black circles indicate the reservoirs used from the Yassin et al. (2019) data set.





**Figure 7.** Mean runoff bias of CLM compared to G-RUN for CONUS and Canada with location of reservoirs (panel a). Simulated (blue line) and observed inflow (black line) seasonality per reservoir (panels b-s)





## 5.4 Global-scale mizuRoute simulations: evaluation for global streamflow indices


We evaluate the global impact of accounting for dam operations on long-term river discharge by comparing the skill of the DAM with the NOLAKES simulation for observed monthly streamflow indices from the GSIM archive (Fig. 8). In general, the DAM simulation shows improved skill compared to the NOLAKES simulation (Fig. 8e-h), with a median absolute percent bias for mean flows of 72 % compared to 81 %. The improvement is particularly strong for the standard deviation, with a mean

absolute percent bias of 187 % for NOLAKES compared to 100 % for DAM, indicating an improvement of the total streamflow variability (Fig. 8c). This is not surprising, as reservoir operations typically minimize streamflow variability (Hanasaki et al., 2006). For high floods, the DAM simulation outperforms NOLAKES (79 % compared to 114 % mean absolute bias), with the best improvements in Canada, Western United States and Central Africa (Fig. 8e). Finally, for low floods, the overall improvement is smaller, with a mean absolute bias of 79 % for DAM compared to 91 % for NOLAKES, with the latter providing

remarkably better results in India and southwestern USA.

Comparing the DAM and NAT simulations, it is remarkable that NAT shows the best skill for monthly standard deviation (Fig. 8f and appendix Fig. A6c), which could point at a better buffering of the biased river streamflow by the natural lake scheme of Döll et al. (2003). This corresponds to the findings of the global-scale mizuRoute evaluation to individual reservoirs

observations (section 5.2). As the Hanasaki et al. (2006) parametrisation mainly depends on mean annual and monthly inflows, it suffers from the inflow biases, while the natural lake parametrisation of Döll et al. (2003) mainly attenuates the incoming inflow. In India and in southeastern US, daily low flows are better represented in the NOLAKES simulation (Fig. 8d). Overall, the difference between NAT and DAM is small compared to the difference between not representing lakes and representing lakes. On average, NAT is outperforming DAM for the mean, standard deviation and monthly maximum indices. For low flows

however, DAM shows the best performance, with a median absolute percent bias of 90 % for the NAT simulation compared to 79 % for DAM (Fig. 8h). Especially in India and in southern Africa, the DAM simulation shows substantially higher skill in representing low flows (appendix Fig. A6).



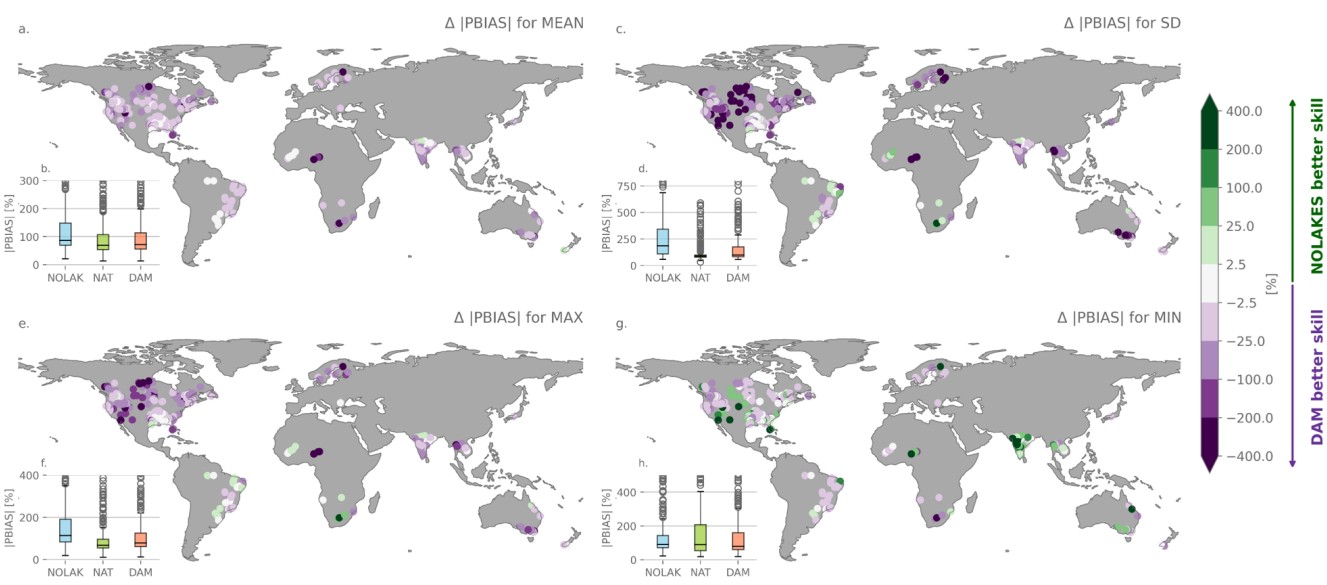

**Figure 8.** Performance of global-scale mizuRoute simulations for streamflow indices of GSIM. Added value in absolute percent bias of accounting for reservoirs (DAM) over simulations without lakes or reservoirs (NOLAKES) ($|PBIAS|_{DAM}$-$|PBIAS|_{NOLAKES}$) for monthly mean streamflow (MEAN; panel a), monthly streamflow standard deviation (SD; panel b), monthly maximum streamflow (MAX; panel c) and monthly minimum streamflow (MIN; panel d). Note the non-linear colorbar scale. Inset panels (e-h) show the $|PBIAS|$ for the simulation without lakes (NOLAKE), with only natural lakes (NAT) and accounting for reservoirs (DAM). Only GSIM stations on the river network, maximum 200 km$^2$ downstream of a reservoir and with observations in the simulation period are included.

# 6 Discussion

## 6.1 Reservoir parametrisation and river network

The parametrisation of Hanasaki et al. (2006) is designed to provide generic operational rules, rather than observation-driven release rules for individual reservoirs (Turner et al., 2020). However, individual calibration could improve simulated releases of modelled reservoirs. Especially for highly regulated rivers with a series of cascading reservoirs, calibration schemes of upstream reservoir releases could improve the modelled river streamflow (Shin et al., 2019). However, prior to conducting such parameter calibration, it would be advisable to first reduce biases in the reservoir inflows as simulated by CLM (section 6.3).

The overestimated inflow for reservoirs with highly regulated upstream flows, like Amistad and Falcon International on the Rio Grande (Fig. 7j-k), is likely due to unresolved reservoirs upstream. For example, only 6 of the 23 dams and water diversions on the Rio Grande are resolved within the current river network, which could be attributed to the following reasons. First, cascade systems and run-of-river dam infrastructures (e.g. Leasburg and Isleta dams on the Rio Grande), which control




the river flow but do not store water, are generally not included in GRanD and therefore are not in the river network. Second, several dams and associated reservoirs are not on the stream network due to the network resolution (e.g. the remote Platoro reservoir on the Conejos river). Third, reservoirs smaller than the area threshold of 10 km$^2$ are not included on the river network (e.g. Sumner reservoir).


These issues could be accommodated by the use of higher resolution stream networks on which more reservoirs would be resolved, like the Multi-Error-Removed-Improved-Terrain (MERIT) Hydro network which is derived from a global DEM at 3 arcsec resolution ($\sim$90 m) (Yamazaki et al., 2019). An accurate high-resolution DEM is important to improve the reservoir representation and release, as has been shown by Shin et al. (2019). The choice of river network proves however to be less

important compared to the runoff input from the land model for global-scale river flow simulations without lakes and reservoirs (Mizukami et al., 2021), so accounting for and reducing runoff biases remains an essential step. Finally, to account for run-of-river dams, the GRanD database could be updated or complemented by other data sources like the Global Georeferenced Database of Dams (GOODD; Mulligan et al., 2020).

**6.2 Irrigation demand and topology**

The local mizuRoute simulations showed only small differences in outflow values for the DAM and DAM_NOIRRIG simulations for irrigation reservoirs. Our results suggest that the total irrigation water demand per reservoir is underestimated, and that there are also potential biases in the irrigation seasonal cycle. These uncertainties are either originating from the irrigation topology, defining the area to which the reservoir water is allocated, or from the gridded irrigation amounts simulated by CLM.

Since applying the irrigation topology with different thresholds (1000 km instead of 700 km downstream along the main river stem and 200 km instead of 100 km along the tributaries) did not significantly improve the irrigation demands, disparities in simulated irrigation amounts likely play a major role.

The irrigation module in CLM is calibrated with one free parameter based on global observed irrigation water withdrawals

from AQUASTAT (Thiery et al., 2017, 2020). It is however possible that these country-based irrigation amounts are under-reported by individual countries. In addition, while global crop calendar data exists to a limited extend (e.g. Sacks et al., 2010), there is almost no information on timing and amount of global irrigation water withdrawals to use for model evaluation. Yet, there are various possible pathways to improve the simulation of irrigation water withdrawal, like differentiating irrigation techniques applied in different regions around the world (Jägermeyr et al., 2015) and including crop rotation and other agricul-

tural management practices (e.g. Hirsch et al., 2017, 2018). Furthermore, the use of remotely sensed soil moisture to estimate the amount and timing of irrigation demonstrates promising results (Brocca et al., 2018; Zaussinger et al., 2019; Massari et al., 2021; Lawston et al., 2017).





Apart from the uncertainties in gridded irrigation demands from CLM, there are several opportunities to improve the ir-
rigation topology routine. Here, we use the HDMA river network topology and derive with simple rules on distance and
bottom elevation of river segments over which HRUs the water demand to a reservoir is aggregated. However, more detailed
river networks, like MERIT-Hydro (Yamazaki et al., 2019) would allow to refine the criteria. For example, it could include
more topological details like the Height Above Nearest Drainage index (Nobre et al., 2011; Gharari et al., 2011). Future im-
provements of the irrigation topology could also account for water transfers, including water diversion for irrigation at weirs
(Hanasaki et al., 2021).

### 6.3 Runoff biases in CLM

The inflow biases shown in section 5.3 for the reservoirs in contiguous United States and Canada, can be roughly subdivided
into reservoirs where there is a bias in inflow timing, and reservoirs where the inflow is largely overestimated, with some excep-
tions. In our modelling framework, the biases in inflow timing for reservoirs with mountainous headwaters could originate from
the lack of a representation of high elevation snow pack, and the associated timing of snow melt, in these relatively coarse reso-
lution simulations. Another potential source of uncertainty is the sensitivity of runoff simulations to the meteorological forcing
providing biased timing and amounts of precipitation, especially in high mountain catchments, which affect the runoff ratios.
We tested these hypotheses by running mizuRoute over the North-American domain using a high resolution CLM simulation
forced with North American Land Data Assimilation System NLDAS meteorological forcing on a high resolution grid (0.125°;
appendix Fig. A5). These simulations did not improve magnitude or timing of the inflow biases, so likely these uncertainties
may be coming from internal dynamics in CLM. For the large reservoirs with headwaters in the plains like Falcon International
and Amistad reservoir, a second reason for the large positive inflow biases next to unresolved upstream river regulation, is
the suspected underestimation of irrigation water amounts applied (section 6.2). In addition, CLM does not include water ab-
stractions for domestic and industrial purposes (Telteu et al., 2021), which would explain the high bias in simulated streamflow.

There are several potential avenues for future model development that could potentially reduce the model runoff and stream-
flow errors, especially the timing errors. Natural processes related to snow accumulation and melt dynamics could be investi-
gated and improved. Ongoing work with the new representative hillslope model within CLM, which now includes temperature
and precipitation downscaling as well as the impacts of slope and aspect on hillslope to lateral flow, could potentially help re-
solve early runoff peak biases (Swenson et al., 2019). Felfelani et al. (2021) have also explored how explicit grid-to-grid lateral
flow can improve high-resolution CLM simulations. Additionally, CLM parameters, which have previously been calibrated for
evapotranspiration and gross primary production (Dagon et al., 2020), could be calibrated for runoff as well.

Another possible pathway to compare the different reservoir parametrisations in global-scale mizuRoute simulations, is to
use input from a high-resolution calibrated hydrological model, like the Variable Infiltration Capacity model (VIC; Liang et al.,
1994; Mizukami et al., 2021) or SWAT (Arnold et al., 2012). As the goal is to have reservoirs represented in a Earth System



modelling framework, this evaluation is out of scope of this study. Another possibility is to use a bias correction on simulated runoff values from CLM using the G-RUN reconstruction, but this did not yield satisfying results.

## 6.4 Towards a representation of reservoirs in a coupled Earth System Model

The modelling framework in this study is a stand-alone application of the routing scheme mizuRoute and the land model CLM, the land component of CESM. The coupling of mizuRoute to CLM and CESM, which is currently ongoing, will enable routing runoff from the land to the ocean with a network-based routing mode, thereby permitting streamflow alteration by dam operations through the reservoir parametrisations. Coupling the vector-based model to the gridded land model requires an on-the-fly remapping step to communicate runoff from the land model to the vector-based river network. As the water balance of natural lakes and reservoirs is simulated within mizuRoute using precipitation and lake evaporation from CLM, the coupling would also enable more realistic lake and reservoir water balance dynamics to the Earth System Model, which were hereto not simulated (Gharari et al., in prep. Vanderkelen et al., 2021; Mizukami et al., 2021).

Moreover, a coupling into CESM would allow us to investigate the direct interactions between land and atmospheric processes, and the dynamics of lakes and reservoirs, including changes in storage and lake/reservoir surface area (Yigzaw et al., 2018). CLM models the energy cycle and atmospheric interaction of lakes as a sub-grid column of the grid cell, which has a constant volume. In a two-way mizuRoute-CLM coupling, the modelled changes in storage, depth and lake area area could be communicated from mizuRoute back to CLM, and would affect the land-climate interactions. A possible pathway is to convert simulated storage into area and depth using the dataset of (Yigzaw et al., 2018), which includes storage-area-depth relationships for the reservoirs in GRanD. CLM already allows for dynamical changes in lakes (Vanderkelen et al., 2021), but some key challenges remain, like the aggregation of the individual reservoirs in mizuRoute to the lake column of the CLM grid cells.

In a coupled framework, not only the runoff will need to be communicated, but also the irrigation water demand and topology. Currently, mizuRoute only uses irrigation demand seasonality to determine the dam release for irrigation reservoirs. A two-way coupling would ultimately allow for water to be extracted directly from the river for irrigation, thereby using runoff generated in upstream grid cells. In this way, the actual availability of water for irrigation would be better represented. To this end, the irrigation topology could serve as a blueprint for transporting irrigation water across grid cells. Eventually, the coupled system will allow to more accurately model the human alteration of water resources globally in the present, and under different future emission and socioeconomic scenarios.

## 7 Conclusions

In this study, we evaluate a reservoir parametrisation (Hanasaki et al., 2006) that we integrated into the river routing model mizuRoute and assess how a simple treatment of human dam regulation affects global streamflow simulations. To this end,

we develop an irrigation topology based on the vector-based river network that provides the area over which water demand
is aggregated for each individual irrigation reservoir. Local mizuRoute simulations for 26 reservoirs using observed inflows
demonstrate that the reservoir parametrisation has added value compared to the natural lake scheme of Döll et al. (2003) for
the simulation of reservoir release and storage. The reservoir parametrisation shows high skill in simulating reservoir storage,
particularly for reservoirs with a multi-year storage capacity. The benefits of accounting for irrigation demand seasonality ap-
pears to be limited in the existing modelling framework, but this could be either due to a spatial sampling bias of reservoirs
with observations available or uncertainties in the simulated irrigation demand.

Biases in modelled river discharge, which can be attributed to runoff biases in CLM, prevent strict validation with obser-
vations of the impact from reservoir operations. However, monthly streamflow indices indicate that accounting for lakes and
reservoir regulation does appear to improve the representation of mean and high flows as well as flow variability, even if the
total amount and timing of runoff is biased.

Our results highlight the opportunities and challenges of global-scale reservoir and streamflow simulations, and provide
an essential step for representing reservoirs in Earth System Models and for incorporating human dam operations in global
assessments of water resources availability under present-day and future climates. Modelling reservoirs in a coupled system
will allow to more accurately evaluate water availability for human consumption, irrigation and ecosystems, and to explore the
role of different reservoir management strategies and priorities in altering water availability under climate change.

*Code and data availability.* The reservoir dataset described in Yassin et al. (2019) is available at http://doi.org/10.5281/zenodo.1492043. The
GSIM data can be found at https://doi.pangaea.de/10.1594/PANGAEA.887477, while the G-RUN ENSEMBLE reconstructions are available
at https://doi.org/10.6084/m9.figshare.12794075. The HydroLAKES dataset is available at https://www.hydrosheds.org/page/hydrolakes,
GRanD at http://globaldamwatch.org/ and the HDMA dataset at https://www.sciencebase.gov/catalog/item/5910def6e4b0e541a03ac98c.
The source code of mizuRoute tag cesm-coupling.n00$_v$2.0.1 $is publicly available at$ and CLM5.0 is available through the Commu-
nity Land Model (CLM) repository: https://github.com/ESCOMP/CLM/. The scripts used in this study are available at: https:
//github.com/VUB-HYDR/2022_Vanderkelen_etal_GMD. Finally, all simulations used in the analysis will become openly
available upon publication.

## Appendix A: Supplementary figures

### A1



**Table A1.** Parameters for the Hanasaki et al. (2006) reservoir parametrisation in MizuRoute.

| Parameter | Unit | Value | Description |
|---|---|---|---|
| $S_{max}$ | m$^3$ | from GRanD | Maximal reservoir storage |
| $\alpha$ | - | 0.85 | Fraction of active storage compared to total storage (value from Hanasaki et al., 2006) |
| $\beta$ | - | 0.9 | Fraction of inflow that can be used to meet demand (value from Biemans et al., 2011) |
| $S_{ini}$ | m$^3$ | $S_{max}$ from GRanD | Initial storage, used to calculate release coefficient before start of operational year |
| $c_1$ | - | 0.1 | coefficient 1 of target release calculation (value from Hanasaki et al., 2006) |
| $c_2$ | - | 0.9 | coefficient 2 of target release calculation (value from Hanasaki et al., 2006) |
| $exponent$ | - | 2 | Exponent in actual release calculation (value from Hanasaki et al., 2006) |
| $denominator$ | - | 0.5 | Denominator in actual release calculation (value from Hanasaki et al., 2006) |
| $c_{compare}$ | - | 0.5 | Criterion to distinguish between "multi-year" and "within-a-year" reservoirs, compared against $c$ (value from Hanasaki et al., 2006) |
| $E_r$ | - | calculated based on GRanD | Release coefficient (provided with initial value and updated every start of operational year) |
| $I_{m,jan}$ - $I_{m,dec}$ | m$^3$ s$^{-1}$ | from CLM (preprocessed) | Mean monthly reservoir inflow |
| $D_{m,jan}$ - $D_{m,dec}$ | m$^3$ s$^{-1}$ | from CLM (preprocessed) | Mean monthly reservoir demand |
| $purpose$ | - | from GRanD | Reservoir purpose (0 non-irrigation, 1 irrigation) |



**Table A2.** Reservoirs of the Yassin et al. (2019) observational dataset used in this study. The asterisk in the observation period column indicates this reservoir has monthly instead of daily observations. Maximum capacity is derived from GRanD.

| Dam name | Country | Main purpose | Capacity (mcm) | Period | capacity ratio |
|---|---|---|---|---|---|
| American Falls | USA | irrigation | 2061.5 | 1978-1995 | 0.30 |
| Amistad | USA/Mexico | irrigation | 6330 | 1977-2002 | 2.48 |
| W. A. C. Bennett | Canada | hydropower | 74300 | 2003-2011 | 3.27 |
| Bhumibol | Thailand | irrigation | 13462 | 1980-1996 | 2.62 |
| Charavak | Uzbekistan | hydropower | 2000 | 2001-2010* | 0.28 |
| Dickson | Canada | water supply | 203 | 2005-2011 | 0.18 |
| E.B. Campbell | Canada | hydropower | 2200 | 2000-2011* | 0.16 |
| Falcon International | USA/Mexico | flood control | 3920 | 1958-2001 | 1.20 |
| Flaming Gorge | USA | water supply | 4336.3 | 1971-2017* | 2.27 |
| Fort Peck | USA | flood control | 23560 | 1970-1999* | 2.43 |
| Garrison | USA | flood control | 30220 | 1970-1999 | 1.41 |
| Ghost | Canada | hydropower | 132 | 1990-2011 | 0.05 |
| Glen Canyon | USA | hydropower | 25070 | 1980-1996* | 1.67 |
| High Aswan | Egypt | irrigation | 162000 | 1971-1997 | 2.79 |
| Navajo | USA | irrigation | 1278 | 1971-2011 | 1.07 |
| Nurek | Tajikistan | irrigation | 10500 | 2001-2010* | 0.50 |
| Oahe | USA | flood control | 29110 | 1970-1999 | 1.22 |
| Oldman | Canada | irrigation | 490 | 1996-2011 | 0.44 |
| Oroville | USA | flood control | 4366.5 | 1995-2004* | 0.72 |
| Palisades | USA | irrigation | 1480.2 | 1970-2000 | 0.24 |
| Seminoe | USA | irrigation | 1254.8 | 1951-2013 | 1.05 |
| Sirikit | Thailand | irrigation | 9510 | 1980-1996 | 1.82 |
| St. Marry | Canada | irrigation | 394.7 | 2000-2011 | 0.50 |
| Toktogul | Kyrgyzstan | hydropower | 19500 | 2001-2010* | 1.39 |
| Trinity | USA | irrigation | 2633.5 | 1970-2000 | 1.51 |
| Yellowtail | USA | irrigation | 1760.6 | 1970-2000 | 0.57 |





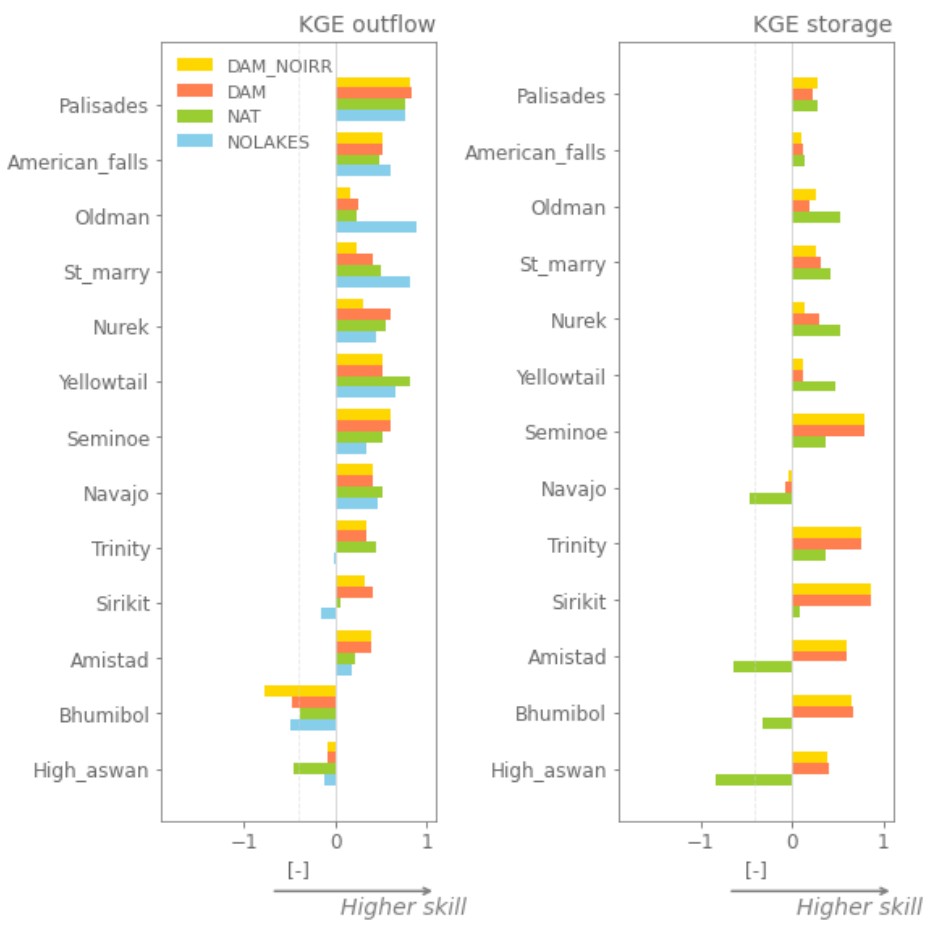

**Figure A1.** Evaluation with Kling-Gupta Efficiency (KGE) for irrigation reservoirs of the Hanasaki et al. (2006) (DAM) and (Döll et al., 2003) (NAT) parametrisations with observed inflows, and run-of-the river conditions (assuming there is no lake; NOLAKE) against observed outflow (panel a) and observed storage (panel b) using observations from Yassin et al. (2019). The reservoirs are ordered from low to high capacity ratio.



**Figure A2.** Time series and seasonal cycles of outflows and storage of observation driven simulations using the Hanasaki et al. (2006) parametrisation with and without accounting for irrigation (DAM and NO_DAM, respectively), the natural lakes Döll et al. (2003) parametrisation (NAT) and run-of-the river conditions (NOLAKES), all for irrigation reservoirs, compared to observations.





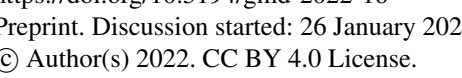

**Figure A2.** Continued.





**Figure A3.** Same as Fig. A2, but for non-irrigation reservoirs.





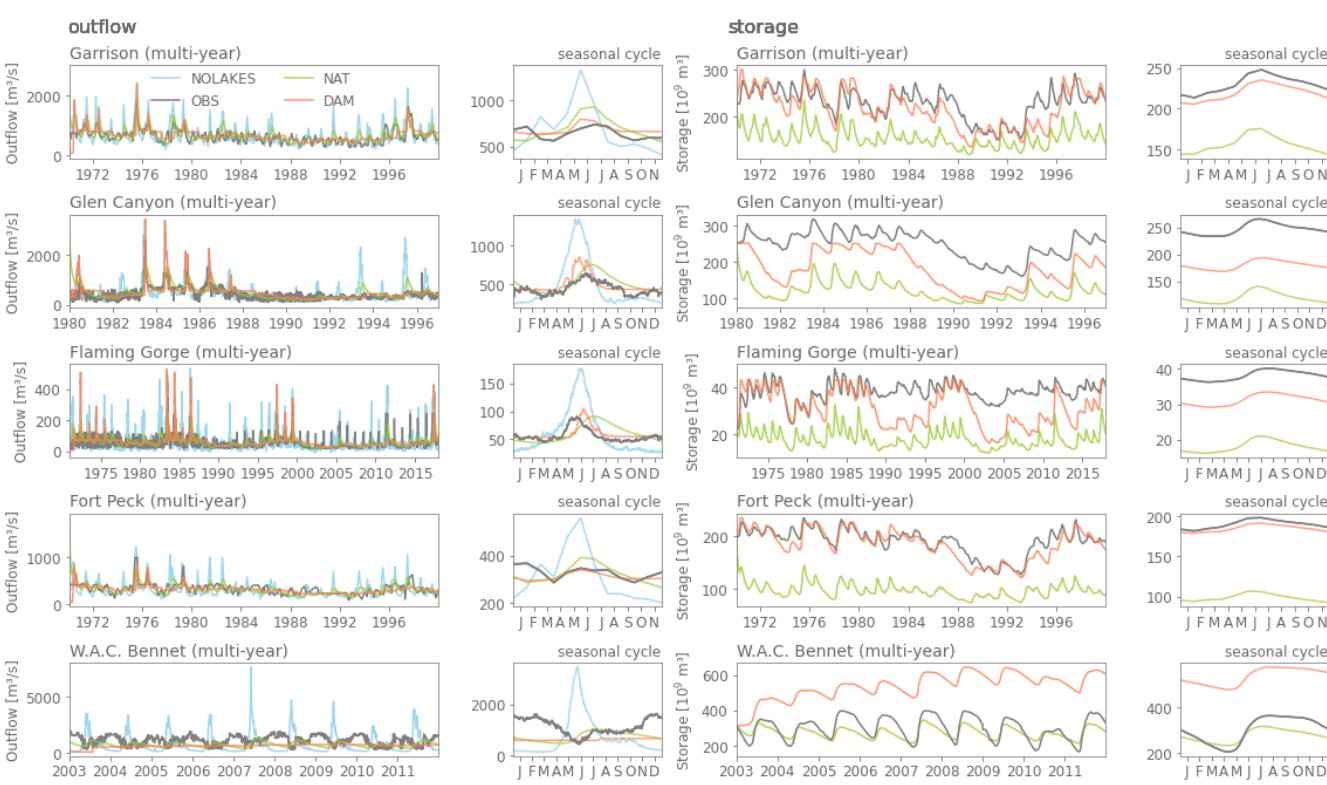

**Figure A4.** Continued.







**Figure A4.** Performance for mizuRoute simulations for outflow (panel a-c) and storage (panel d-f) compared to reservoir observations using the KGE terms: variability error ($\frac{\sigma_{mod}}{\sigma_{obs}}$; panels a, c), mean bias ($\frac{\mu_{mod}}{\mu_{obs}}$; panels b, e) and correlation (panels c, f).



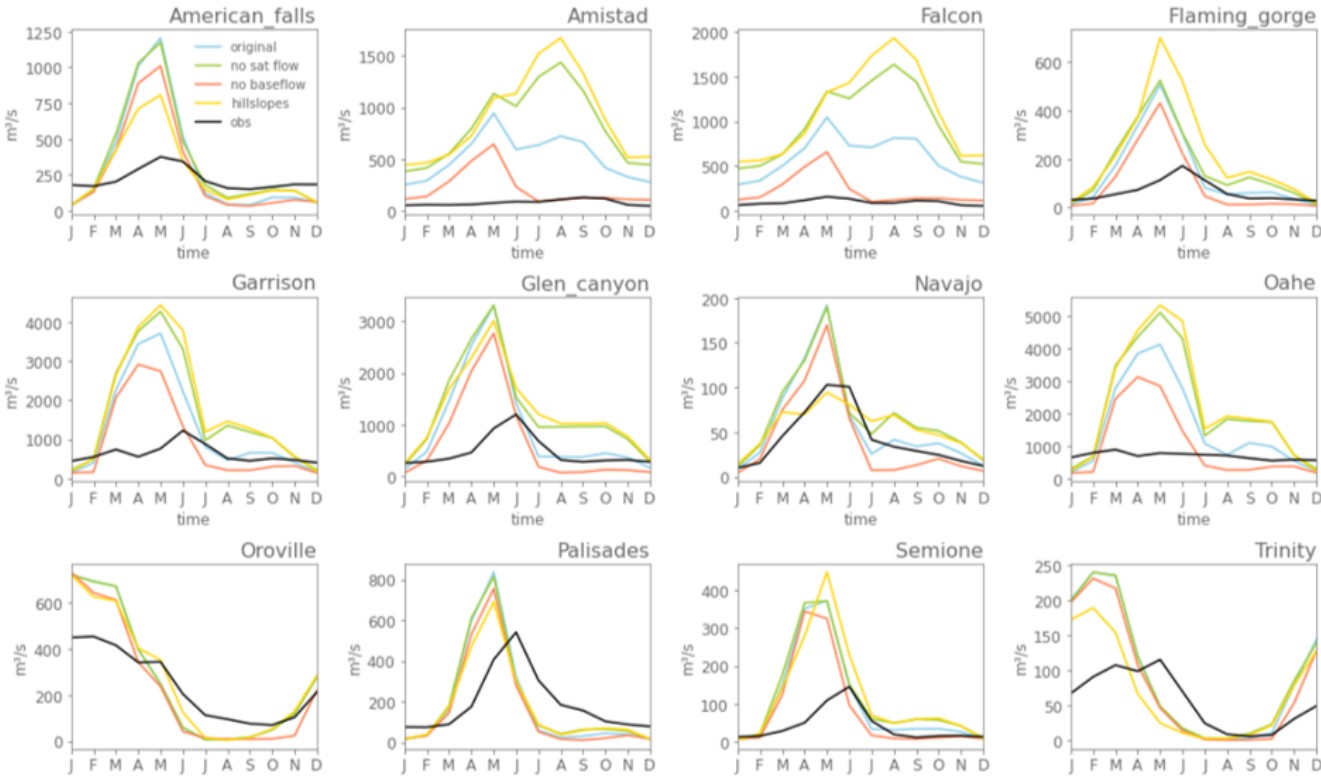

**Figure A5.** Simulated and observed inflow seasonality per reservoir with mizuRoute using runoff from different CLM simulations at 0.125°
resolution with meteorological forcing from NLDAS. In the legend, 'original' refers to the simulation with the default CLM version used
in the main analysis, 'no sat flow' refers to the simulation where surface saturation excess runoff is set to 0, 'no baseflow' refers to the
simulation with a decreased baseflow parameter, 'hillslopes' refers to the simulation using the hillslope model described in Swenson et al.
(2019), performed at 0.5° horizontal resolution, and 'obs' are the observed inflows from the Yassin et al. (2019) dataset.



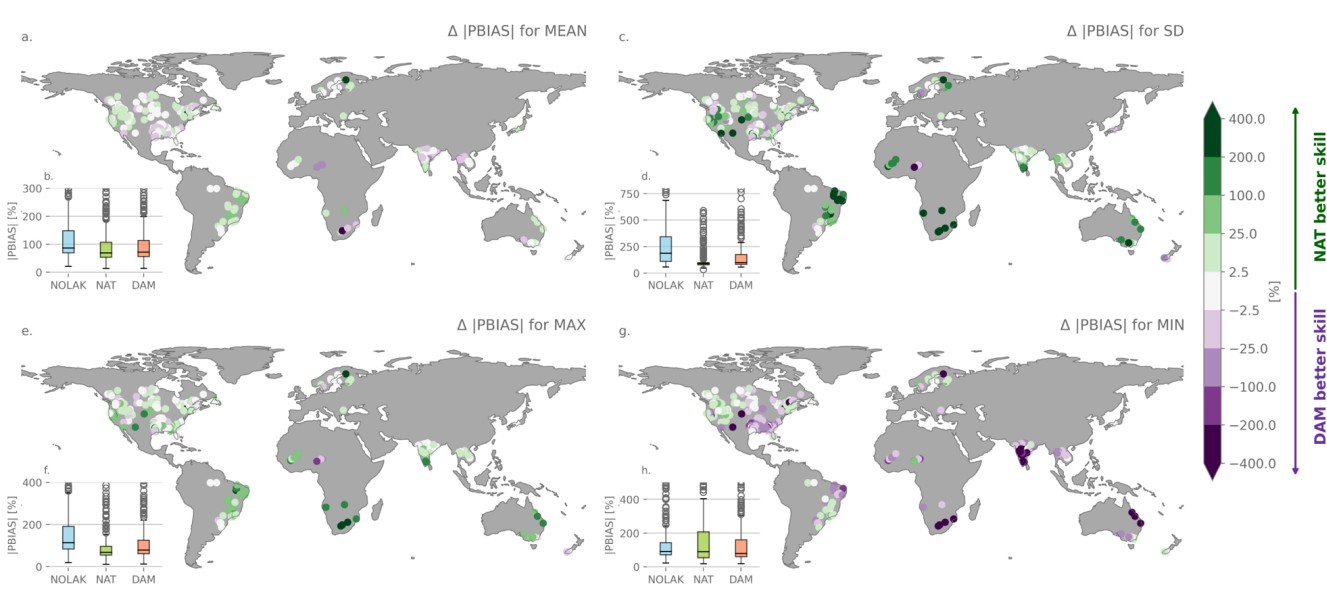

**Figure A6.** Same as Fig. 8 but for the absolute percent bias for natural lakes compared to reservoirs ($|PBIAS|_{DAM}-|PBIAS|_{NAT}$)



*Author contributions.* IV, SH and NM implemented the reservoir parametrisation in mizuRoute. IV performed and analysed the simulations. SS, DML and MC provided scientific input on CLM and runoff biases. MC, DML, YP, NH, AvG and WT provided general oversight and guidance. IV wrote the manuscript with input from all co-authors.

*Competing interests.* The authors declare there are no competing interests.

*Acknowledgements.* The authors would like to thank Erik Kluzek for his support on CLM. Inne Vanderkelen is a research fellow at the Research Foundation Flanders (FWOTM920). The CESM project is supported primarily by the National Science Foundation (NSF). This material is based upon work supported by the National Center for Atmospheric Research, which is a major facility sponsored by the NSF under Cooperative Agreement No. 1852977. Computing and data storage resources, including the Cheyenne supercomputer (doi:10.5065/D6RX99HX), were provided by the Computational and Information Systems Laboratory (CISL) at NCAR. We thank all the scientists, software engineers, and administrators who contributed to the development of CESM2. Other storage resources and services used in this work were provided by the VSC (Flemish Supercomputer Center), funded by the Research Foundation - Flanders (FWO) and the Flemish Government. This study was supported by the LAMACLIMA project, part of AXIS, an ERA-NET initiated by JPI Climate, and funded by BELSPO (BE, Grant No. B2/181/P1) with co-funding by the European Union (Grant No. 776608).



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
