# Peer review of "Evaluating a reservoir parametrisation in the vector-based global routing model mizuRoute (v2.0.1) for Earth System Model coupling"

_Geoscientific Model Development, 2022_

## Author Response (AR1)

**Evaluating a reservoir parametrisation in the vector-based global routing model mizuRoute (v2.0.1) for Earth System Model coupling**

Geoscientific Model Development

May 2, 2022

I. Vanderkelen1, S. Gharari2, N. Mizukami3, M. Clark2 D. M. Lawrence3, S. Swenson 3 Y. Pokhrel4, N. Hanasaki5, A. van Griensven1, and W. Thiery1 inne.vanderkelen@vub.be

1 Vrije Universiteit Brussel, Department of Hydrology and Hydraulic Engineering, Brussels, Belgium 2 University of Saskatchewan, Centre for Hydrology and Coldwater Laboratory, Canmore, Canada 3 National Center for Atmospheric Research, Boulder, Colorado, USA 4 Michigan State University, Department of Civil and Environmental Engineering, East Lansing, MI, United States

5 National Institute for Environmental Studies, Tsukuba, Japan

**Contents**

| 1 | Reviewer 1             | 3  |
|---|------------------------|----|
|   | 1.1 Major comments     | 3  |
|   | Reviewer 1 Comment 1   | 3  |
|   | Reviewer 1 Comment 2   | 3  |
|   | Reviewer 1 Comment 3   | 7  |
|   | 1.2 Minor comments     | 8  |
|   | Reviewer 1 Comment 4   | 8  |
|   | Reviewer 1 Comment 5   | 9  |
|   | Reviewer 1 Comment 6   | 9  |
| 2 | Reviewer 2             | 11 |
|   | 2.1 Major comments     | 11 |
|   | Reviewer 2 Comment 1   | 11 |
|   | Reviewer 2 Comment 2   | 11 |
|   | Reviewer 2 Comment 3   | 11 |
|   | 22 Specific comments   | 12 |
|   | Reviewer 2 Comment 4   | 12 |
|   | Portiouror 2 Commont 5 | 12 |
|   |                        | 13 |
|   | 2.3 Figure comments    | 14 |
|   | Reviewer 2 Comment 6   | 14 |
|   | Reviewer 2 Comment 7   | 14 |

| 2.4 Technical comments | 19 |
|------------------------|----|
| Reviewer 2 Comment 8   | 19 |
| Reviewer 2 Comment 9   | 19 |
| Reviewer 2 Comment 10  | 19 |
| Reviewer 2 Comment 11  | 19 |
| Reviewer 2 Comment 12  | 20 |
| Reviewer 2 Comment 13  | 20 |

**Abstract**

This response letter contains numbered figures and references to these figures. To prevent confusion, the figures embedded within this response letter are called illustrations. Finally, the following convention is applied to denote modification in the original manuscript: new text.

**1 Reviewer 1**

**1.1 Major comments**

**Reviewer 1 Comment 1**

Vanderkelen et al. present an analysis of reservoir storage implementation in mizuRoute. The paper is very well written with excellent level of detail in method description and clear results. Results are unsurprising and lead to little in the way of new insight from a pure science perspective. The paper is therefore appropriately targeted to GMD rather than a research-oriented journal. I recommend publication if just a couple of relatively minor issues and omissions of recent data/literature can be addressed.

**Response**

We thank Reviewer 1 for the overall support of the study and the constructive feedback to improve the manuscript. Below, we address every comment carefully and explain the corresponding changes in the manuscript.

**Reviewer 1 Comment 2**

Just 26 sites are used to test the performances of the various model settings applied. This is far too few for a robust analysis of model performance. The authors will be interested in the recent data publication by Steyaert et al. (2022), which provides daily storage and flows for approximately 700 dams in the US - https://www.nature.com/articles/s41597-022-01134-7. Although these data are US centric, they will provide a far better sample for performance analysis.

**Response**

We are grateful to Reviewer 1 for pointing us to the ResOpsUS dataset of Steyaert et al. (2022), however we would like to mention that this paper has been published on February 3rd, which is after the initial submission of our manuscript (18th of January). Still, we extended the evaluation of the global mizuRoute simulations with individual reservoir observations of this new dataset. To this end, we used a subset of 32 reservoirs which all have both storage and outflow observations available, and are resolved on the HDMA river network.

The figures with absolute PBIAS and KGE values for outflow (Illustration 1) and storage (Illustration 2) confirm the results of the 26 reservoirs from the Yassin et al. (2019) dataset and are included in the appendix. However, since the ResOpsUS dataset only contains reservoirs within the contiguous United States, this dataset does not elevate the spatial sampling bias of reservoir observations related to irrigation reservoirs mentioned in the conclusion of the manuscript.

Finally, as 26 individual reservoirs are too little, in the original version of the manuscript we

also analyse streamflow indices using the GSIM dataset (Section 5.4 and Fig. 8) on a global scale.

In addition to the new appendix figures, we added the following text to the data and result sections of the manuscript:

**4.1 Local reservoir observations**

[...] To evaluate the global mizuRoute simulations, we complement the reservoir observations from Yassin et al. (2019), with the ResOpsUs historical reservoir data set for the contiguous United States (Steyaert et al., 2022). Of the 679 reservoirs in the dataset, we use a subset of 32 reservoirs for which both outflow and storage observations are available within the simulation period, and that are resolved on the employed HDMA river network.

**5.2 Global-scale mizuRoute simulations: evaluation with reservoir observations**

[...] Consistent with the observation-driven local simulations, the global-scale DAM simulation performs systematically better for reservoirs with a high capacity ratio, and in most cases better than NAT. These findings are generally confirmed by the evaluation with the ResOpsUS reservoir observations, where the DAM outperforms the NAT simulation for 13 of the 32 reservoirs (Fig. A5).

[...] The same pattern is found when comparing simulated storage to the observed storage from the ResOpsUS dataset (Fig. A6). The next section therefore focuses on the biases in simulated inflow and runoff.

Higher skill

Higher skill

Illustration 1: Performance of the global-scale mizuRoute simulations for outflow compared to reservoir observations from the ResOpsUs dataset (Steyaert et al., 2022) using absolute percent bias (|PBIAS|; panel a) and Kling-Gupta Efficiency (KGE; panel b)